# Causal Heat Conduction Contravening the Fading Memory Paradigm

**DOI:** 10.3390/e21100950

**Published:** 2019-09-28

**Authors:** Luis Herrera

**Affiliations:** Instituto Universitario de Física Fundamental y Matematicas, Universidad de Salamanca, 37007 Salamanca, Spain; lherrera@usal.es

**Keywords:** causal dissipative theories, fading memory paradigm, relativistic dissipative theories

## Abstract

We propose a causal heat conduction model based on a heat kernel violating the fading memory paradigm. The resulting transport equation produces an equation for the temperature. The model is applied to the discussion of two important issues such as the thermohaline convection and the nuclear burning (in)stability. In both cases, the behaviour of the system appears to be strongly dependent on the transport equation assumed, bringing out the effects of our specific kernel on the final description of these problems. A possible relativistic version of the obtained transport equation is presented.

## 1. Introduction

In the study of dissipative processes, the order of magnitude of relevant time scales of the system (in particular the thermal relaxation time) play a major role, due to the fact that the interplay between this latter time scale and the time scale of observation, critically affects the obtained pattern of evolution. This applies not only for dissipative processes out of the steady–state regime but also when the observation time is of the order of (or shorter than) the characteristic time of the system under consideration. Besides, as has been stressed before [1], for time scales larger than the relaxation time, transient phenomena affect the future of the system, even for time scales much larger than the relaxation time.

Starting with the original Maxwell works, dissipative processes were initially studied by means of parabolic theories. These have been proved very useful especially in the steady–state regime [2].

In the context of these theories, the well known classical Maxwell–Fourier law for heat current reads
(1)$$\vec{q}=-\kappa \vec{\nabla }T\;,$$
where κ, *T* and $\vec{q}$ denote the heat conductivity of the fluid, the temperature and the heat flux vector respectively. Using the continuity equation
(2)$$\frac{\partial u}{\partial t}=-\vec{\nabla }.\vec{q},$$
and the constitutive equation for the internal energy *u*
(3)du=γdT,
we obtain the parabolic equation for temperature (diffusion equation),(4)$$\;\frac{\partial T}{\partial t}=\chi \nabla^{2}T,$$
(where $\chi \equiv \frac{\kappa }{\gamma }$ and γ are the diffusivity, and heat capacity per volume, respectively), which, as is well known from the theory of parabolic differential equation, does not describe a causal propagation of perturbations. In other words, perturbations propagate with infinite speed.

This lack of causality is at the origin of alternative proposals based on hyperbolic theories of dissipation [3,4], describing the propagation of perturbations with finite speed.

These causal theories include the dissipative fluxes as field variables, are supported by statistical fluctuation theory and kinetic theory of gases and, furthermore, conform, to a high degree, with experiments [5].

In these theories a fundamental concept is that of the relaxation time τ of the corresponding dissipative process. This positive–definite quantity measures the time required by the system to return to the steady state, once it has been abandoned. It must be emphasized that although it may be sometimes connected to the mean collision time $t_{c}$ of the particles responsible of the dissipative process, it should not be identified with it. Indeed, not only are these two time scales different but, still worse, there is no general formula linking τ and $t_{c}$.

The origin of the pathological behavior exhibited by (Equation 1) stems from the fact that the Maxwell–Fourier law, assuming explicitly the vanishing of the relaxation time, implies that the heat flow starts (vanishes) simultaneously with the appearance (disappearance) of a temperature gradient.

However, even though τ is very small for many processes such as phonon–electron and phonon–phonon interaction at room temperature, ($O(10^{-11})$ and $O(10^{-13})$ seconds, respectively [6]), neglecting it produces the acausality mentioned above and lead to unphysical predictions as for example in superfluid Helium [7,8] and degenerate stars where thermal conduction is dominated by electrons.

In order to overcome this problem a generalization of the Maxwell–Fourier law was proposed by Cattaneo [9,10] and (independently) Vernotte [11], by assuming a non–vanishing thermal relaxation time. The corresponding equation reads

(5)$$\tau \frac{\partial \vec{q}}{\partial t}+\vec{q}=-\kappa \vec{\nabla }T.$$

This equation (known as Cattaneo-Vernotte’s equation) leads to a hyperbolic equation for the temperature (telegraph equation)
(6)$$\tau \frac{\partial^{2}T}{\partial t^{2}}+\frac{\partial T}{\partial t}=\chi \nabla^{2}T\;,$$
which describes the propagation of thermal signals with a finite speed(7)$$v=\sqrt{\chi /\tau }.$$

From all the above, it should be already obvious that the relaxation time cannot be neglected during the transient regimes. Furthermore, it is also obvious that for time scales of the order of (or smaller than) the relaxation time, neglecting the relaxation time is equivalent to disregard the whole process under consideration.

In the past it has been argued that hyperbolic theories may not be necessary after all. The rationale behind this (wrong) conclusion is that when relaxation times are comparable to the characteristic time of the system, it is out of the hydrodynamic regime. Such argument is fallacious, and goes as follows—in the hydrodynamic regime (which we assume) the ratio between the mean free path of fluidparticles and the characteristic length of the system must be lower that unity, otherwise the regime becomes Knudsen’s. In this latter case the material cannot longer be considered as a fluid, in the usual sense. The fact that sometimes (though not always), $t_{c}$ and τ may be of the same order of magnitude, might lead to the erroneous conclusion that “large” relaxation times implies that the system is no longer in the hydrodynamic regime.

But that conclusion is valid only in the case fluidparticles (the ones making up the fluid) and the ones that transport the heat are the same. However this is (almost) never the case. For example, for a neutron star, τ is of the order of the scattering time between electrons (which carry the heat), nevertheless we consider that the neutron star is formed by a Fermi fluid of degenerate neutrons. The same can be said for the second sound in superfluid Helium and solids and for almost any ordinary fluid. To summarize, the hydrodynamic regime refers to fluidparticles, which not necessarily (and as a matter of fact, almost never) are responsible for the heat conduction. Accordingly, large relaxation times (large mean free paths of particles involved in heat transport) do not imply a violation of the hydrodynamic regime (this fact is often overlooked, see Reference [12] for a more detailed discussion on this point).

Finally, it is worth mentioning that problems of the kind we analyze here, have recently attracted the attention of researchers working on fractional calculus (see References [13,14] and references therein). Thus, it would be interesting to find out what new insights, on the problem under consideration here, could be obtained by means of such methods.

## 2. Thermal Memory and Integral Representation of the Heat Flux Vector

It is useful to write heat transport equation such as (Equation 5) in an integral form. The idea behind such an approach comes from the theory of non–linear materials for which the stress at a point at any time is determined by deformation gradients not only at that instant but also at previous instants (see References [15,16,17,18]).

For dissipative processes, integral representations of heat fluxes appear in early works of Coleman, Gurtin and collaborators [19,20,21].

Thus, solving (Equation 5) for $\vec{q}$ we obtain
(8)$$\vec{q}=-\frac{\kappa }{\tau }\int_{-\infty }^{t}\exp\left[-\frac{\left(t-t^{\prime }\right)}{\tau }\right]\cdot \vec{\nabla }T\left(\vec{x},t^{\prime }\right)dt^{\prime }.$$

Obviously, taking the *t*-defivative of (Equation 8) we get (Equation 5).

The above equation is a particular case of the most general expression
(9)$$\vec{q}=-\int_{-\infty }^{t}Q\left(t-t^{\prime }\right)\vec{\nabla }T\left(\vec{x},t^{\prime }\right)dt^{\prime },$$
where the kernel *Q* describes the thermal memory of the material, its role consisting in distributing the relevance of temperature gradients at different moments in the past. Also, it should be mentioned that in the expression above (as well as in all integrals expressions below) the integral domain selected is (−∞,t], instead of [0,t], doing so we are implicitly neglecting the initial effects.

Thus assuming
(10)$$Q=\kappa \delta (t-t^{\prime }),$$
we obtain
(11)$$\vec{q}=-\kappa \vec{\nabla }T\;\left(\mathrm{Maxwell}-\mathrm{Fourier}\right).$$

In other words the Maxwell–Fourier law corresponds to a zero–memory material for which the only relevant temperature gradient is the “last" one, that is, the one simultaneous with the appearance of $\vec{q}$.

On the other hand if we assume
(12)Q=β=constant,
we obtain
(13)$$\frac{\partial^{2}T}{\partial t^{2}}=\frac{\beta }{\gamma }\nabla^{2}T.,$$
where β is a constant with units $\frac{\left[\kappa \right]}{\left[t\right]}$. The above case corresponds to a material with infinite memory, leading to an undamped wave with velocity $v=\sqrt{\frac{\beta }{\gamma }}$.

This particular case describes propagation of thermal waves without attenuation, which is physically objectionable. Furthermore the integral (Equation 9) would diverge. To avoid this drawback and to assure convergence, it might be convenient, for this particular case, to write instead of (Equation 9),

(14)$$\vec{q}\left(x,t\right)-\vec{q}\left(x,t=0\right)=-\int_{0}^{t}Q\left(t-t^{\prime }\right)\vec{\nabla }T\left(\vec{x},t^{\prime }\right)dt^{\prime }.$$

However, for the applications analyzed here, the convergence problem is not relevant and therefore we shall use (Equation 9).

In this context, the Cattaneo-Vernotte equation appears as compromise between the two extreme cases considered above and for which all temperature gradients contribute to $\vec{q}$ but their relevance diminishes as we go farther to the past, as is apparent from its corresponding kernel:(15)$$Q=\frac{\kappa }{\tau }e^{-(t-t^{\prime })/\tau }.$$

Thus, we can say that the temperature gradients in the “neighborhood” of the time *t* provide the main contribution for the heat flux vector appearing in t+τ. Then, assuming that τ is sufficiently small, we may expand $\vec{q}$ around *t* in power series of τ and keep only linear terms, whereas the integral in (Equation 8) may be written approximately as $-\kappa \vec{\nabla }T\left(\vec{x},t\right)$. Combining these two results, Equation (Equation 8) becomes (Equation 5).

The decreasing in the relevance of older temperature gradients as compared with the newer ones is referred to as the “fading memory” paradigm, in the literature. Once again, this basic assumption was introduced initially in the general theory of materials to express the idea that the recent history of deformation should have a greater effect than the remote one, on the present value of the stress [18,22,23]. The extension of this assumption to the heat kernel [19,20,21,24,25] is straightforward and seems to fit intuitively well with “common sense”. Thus the kernels corresponding to the Cattaneo and the Maxwell–Fourier laws, satisfy the “fading memory” paradigm, whereas the kernel (Equation 12) does not.

However, no matter how intuitively acceptable the idea expressed by the “fading memory” paradigm may be, the fact remains that there is not a compelling reason to exclude heat kernels not complying with this paradigm for some possible physical situations and therefore, we are perfectly legitimated to explore the possible existence of heat kernels not satisfying the “fading memory” paradigm and to analyze their potential applications to different physical scenarios. The next sections are devoted to this endeavour.

## 3. Violating the Fading Memory Paradigm

We shall now consider the possibility that, for reasons that will be discussed latter, the material under consideration is such that the heat flux vector depends stronger on the older temperature gradients than on the newer ones. In other words we shall consider a material whose thermal memory behaves all the opposite as expected from the fading memory paradigm. The resulting model will be applied to two important physical issues, namely, the thermohaline convection and the (in)stability of nuclear burning.

### 3.1. The kernel and the Equation for the Temperature

Thus, let us consider a thermal kernel of the form:(16)$$Q=c[1-e^{-(t-t^{\prime })/\tau }],$$
where *c* is a constant with units $\frac{\left[\kappa \right]}{\left[t\right]}$. For simplicity we choose $c\equiv \frac{\kappa }{\tau }$.

Feeding back this kernel into (Equation 9), we obtain for the heat flux
(17)$$\vec{q}=-\frac{\kappa }{\tau }\int_{-\infty }^{t}\left[1-e^{-(t-t^{\prime })/\tau }\right]\vec{\nabla }T\left(\vec{x},t^{\prime }\right)dt^{\prime }.$$

It is clear from the above expression that the oldest temperature gradients have more influence on $\vec{q}$, that the present one. More so, the strongest influence comes from the gradient at infinite past ($t^{\prime }\to -\infty$), whereas the gradient at $t=t^{\prime }$ (the present time) is irrelevant, thereby contradicting the “fading memory paradigm”.

Next, using the Leibniz rule it is a simple matter to find from (Equation 17),
(18)$$\frac{\partial \vec{q}}{\partial t}=-\frac{\kappa }{\tau^{2}}\int_{-\infty }^{t}e^{-(t-t^{\prime })/\tau }\vec{\nabla }T\left(\vec{x},t^{\prime }\right)dt^{\prime },$$
producing
(19)$$\tau \frac{\partial \vec{q}}{\partial t}+\vec{q}=-\frac{\kappa }{\tau }\int_{-\infty }^{t}\vec{\nabla }Tdt^{\prime }.$$

The physical meaning of this last equation, becomes intelligible when we recall that in our model, unlike the Cattaneo–Vernotte equation, the temperature gradients in the neighborhood of *t* are irrelevant and the most important contributions to $\vec{q}$ come from the remote past ($t^{\prime }\to -\infty$).

From (Equation 19), different equations for the temperature may be obtained, depending on the continuity equation to be assumed. This is a sensitive issue, since, in general, the continuity equation depends on the transport law (see Reference [26,27] for a discussion on this point).Therefore, for any specific application of the resulting equation for the temperature, this issue must be handled with some care. Nevertheless, in the applications to be considered in the following sections, the explicit equation for the temperature wont be necessary.

As an example, for simplicity, we shall consider here the standard case, which might not be compatible with the proposed transport law. Thus, combining (Equation 19) with the continuity Equation (Equation 2) for the internal energy *u*, we obtain
(20)$$\frac{\partial^{2}T}{\partial t^{2}}+\frac{1}{\tau }\frac{\partial T}{\partial t}=\frac{\kappa }{\tau^{2}\gamma }\int_{-\infty }^{t}{\vec{\nabla }}^{2}Tdt^{\prime },$$
or, taking the *t*-derivative of the above equation, we obtain the third order equation:(21)$$\frac{\partial^{3}T}{\partial t^{3}}+\frac{1}{\tau }\frac{\partial^{2}T}{\partial t^{2}}=\frac{\kappa }{\tau^{2}\gamma }{\vec{\nabla }}^{2}T.$$

### 3.2. Applications

In the past, different hyperbolic transport equations have been used to study convection processes [28,29,30,31,32,33,34] and in general a variety of different interesting physical phenomena [35,36,37,38,39]. In what follows we shall consider two important problems involving dissipative processes, adopting our kernel (Equation 16).

#### 3.2.1. Thermohaline Instability

Thermohaline convection may be observed in oceans whenever a layer of warm salt water is above a layer of fresh cold water. As far as the salt water is warm enough as to reduce its specific weight to below that of the fresh water, the system is dynamically stable. Thus, if a blob of the upper layer is pushed downward, buoyancy will push it back. However, the cooling of the warm salt water leads to an increasing of its density, decreasing thereby the buoyancy, producing eventually the sinking of small blobs of salty water (the so called “salt fingers”). The interesting point is that instabilities of this kind can also occur in stars, under a variety of circumstances [40].

This kind of secular instability is controlled by the heat leakage of the blob, and therefore the sinking velocity of the blob critically depends on the heat transport equation used to calculate it.

The general expression for the sinking velocity of the convective blob reads (see Reference [40] for details):(22)$$V_{sink.}=-\frac{H_{p}}{\left(\nabla_{ad}-\nabla \right)\tau_{d}}\frac{DT}{T}$$
where $\tau_{d}$ denotes the thermal adjustment time and DT is the difference of the temperature between the convective blob and the surroundings. $H_{P}$ is the scale height of pressure, defined as
(23)$$H_{P}=-P\frac{dr}{dP},$$
where *P* is the pressure and *r* the spatial coordinate along the sinking direction of the blob. Finally, ∇ measures the variation of the temperature with the pressure of the surroundings, whereas $\nabla_{ad}$ measures the variation of the temperature with pressure of the blob, at constant entropy (*E*), that is,
(24)$$\nabla \equiv \left(\frac{d\ln T}{d\ln P}\right),\;\nabla_{ad}\equiv {\left(\frac{\partial \ln T}{\partial \ln P}\right)}_{E}.$$

As mentioned above, the critical point in the calculation of $V_{sink.}$ is the calculation of DT(t).

Thus, let us consider a spherical convective blob with diameter *d* and temperature $T_{b}$. Denoting by $T_{s}$ the temperature of the surrounding fluid, then $DT=T_{b}-T_{s}.$ Approximating the temperature gradient by |∇T|≈2DT/d we obtain for the energy loss λ per unit of time from the whole surface *S* of the blob:(25)$$\lambda =S|\vec{q}|=\frac{2S}{d}\int_{-\infty }^{t}Q\left(t-t^{\prime }\right)DT\left(t^{\prime }\right)dt^{\prime }.$$

It is worth mentioning that the energy loss may be due to thermal conduction as well as radiation, for which we use the diffusion approximation.

On the other hand we know from thermodynamics that the rate by which the thermal energy of the blob of volume *V* is lost (λ), may be written as
(26)$$\lambda =-\rho c_{P}V\frac{\partial T_{b}}{\partial t}\approx -\rho c_{P}V\frac{dDT}{dt},$$
where $c_{P}$ and ρ denote the specific heat at constant pressure and the mass density respectively.

Equating (Equation 25) and (Equation 26) we obtain:(27)$$\frac{dDT}{dt}=-\frac{1}{\kappa \tau_{d}}\int_{-\infty }^{t}Q\left(t-t^{\prime }\right)DT\left(t^{\prime }\right)dt^{\prime },$$
where $\tau_{d}\equiv \frac{\tau d^{2}\rho c_{P}}{12k}$.

Therefore, from (Equation 22) different velocity profiles will be obtained depending on the specific description of the energy loss of the convective blob (i.e., the specific form of the kernel $Q(t-t^{\prime })$).

Thus, if we use the kernel (Equation 10) corresponding to the Maxwell–Fourier law, (Equation 27) becomes
(28)$$\frac{dDT}{dt}=-\frac{1}{\tau_{d}}DT\left(t\right),$$
whose simple solution reads
(29)$$DT=DT\left(0\right)e^{-\frac{t}{\tau_{d}}},$$
that is, we have a monotonically cooling of the blob, with the e-folding time defined by the thermal adjustment time, as expected from purely physical considerations if we neglect the relaxation time.

However, if instead of using (Equation 10), we use (Equation 12) then one obtains from (Equation 27):(30)$$\frac{dDT}{dt}=-\frac{\beta }{\kappa \tau_{d}}\int_{-\infty }^{t}DT\left(t^{\prime }\right)dt^{\prime },$$
or
(31)$$\frac{d^{2}DT}{dt^{2}}+\frac{\beta }{\kappa \tau_{d}}DT\left(t\right)=0,$$
which describes an undamped harmonic oscillation with frecuency $\omega =\sqrt{\frac{\beta }{\kappa \tau_{d}}}$.

The thermohaline instability using the kernel (Equation 15) was studied in Reference [41]. In this case the Equation (Equation 27) for DT becomes
(32)$$\frac{d^{2}DT}{dt^{2}}+\frac{1}{\tau }\frac{dDT}{dt}+\frac{1}{\tau \tau_{d}}DT\left(t\right)=0,$$
with a solution of the form:(33)$$DT=DT\left(0\right)e^{\alpha t}\cos\omega t,$$
describing a damped oscillation with frequency $\omega =\frac{1}{2\tau }\sqrt{\frac{4\tau -\tau_{d}}{\tau_{d}}}$ and damping factor $\alpha =-\frac{1}{2\tau }$. Thus, as it sinks the convection blob oscillates, with a decreasing amplitude. The physical (observational) consequences of this behaviour have been discussed in some detail in Reference [41].

The oscillatory behaviour of the blob before relaxation, observed in this case, brings out the relevance of the specific transport equation to be used in each situation and the richness of the physical phenomena hidden behind the assumption of a vanishing relaxation time.

Let us now analyze the problem of the thermohaline convection with our kernel (Equation 16). Proceeding as for the previous examples, we obtain for DT

(34)$$\frac{d^{3}DT}{dt^{3}}+\frac{1}{\tau }\frac{d^{2}DT}{dt^{2}}+\frac{1}{\tau^{2}\tau_{d}}DT\left(t\right)=0.$$

This is a homogeneous linear, third order differential equation whose general properties are well known (see, for example, Reference [42]).

In general for an equation of the form
(35)$$\frac{d^{3}Y}{dt^{3}}+a\frac{d^{2}Y}{dt^{2}}+b\frac{dY}{dt}+cY=0,$$
where a,b,c are real constants, it can be shown that the solution is of the form (see pages 44–47 in Reference [42])
(36)$$Y=e^{\alpha t}\left(\cos\omega t+\sin\omega t\right),$$
if and only if
(37)$$\frac{2a^{3}}{27}-\frac{ab}{3}+c-\frac{2}{3\sqrt{3}}{\left(\frac{a^{2}}{3}-b\right)}^{3/2}>0.$$

In our case $a\equiv \frac{1}{\tau }$, b=0 and $c\equiv \frac{1}{\tau^{2}\tau_{d}}$, implying that the above inequality is always satisfied. The explicit form of the solution requires to solve the characteristic equation corresponding to (Equation 34). This is a cubic algebraic equation whose roots are given by rather cumbersome and not very illuminating expressions at this point. Suffice is to say that we have in this case also a damped oscillatory motion, although the frequency and the damping factor are different from the ones obtained with the Cattaneo–Vernotte equation.

#### 3.2.2. Secular Stability of Nuclear Burning before Relaxation

The secular stability of nuclear burning is an issue of utmost interest in astrophysics. Indeed, early work by Rosenbluth et al. [43] shows how the hydrogen falling onto the surface of a neutron star in a close binary system undergoes nuclear fusion, independently on how low the temperature may be. This idea has been exploited by many authors for the modelling of compact x–ray sources (see References [44,45,46] and references therein).

The nuclear instability reported in the above references, appears whenever the characteristic time for the increasing of the thermal energy generated by nuclear burning is smaller than the time required for the removal of this energy. Also, as shown by Hansen and Van Horn [44], the range of time scales for growth of the instability span from milliseconds to minutes. Therefore, since the relaxation time may be of the order of milliseconds (or larger) for highly degenerate matter, it appears evident that the Maxwell–Fourier law which implies the vanishing of the relaxation time, might not be appropriate to describe this kind of problem and we need to resort to transport equations including the relaxation time.

Two main ideas are required for our discussion. One is the concept of gravothermal specific heat and the other is the equation relating the fluctuations of the energy released by nuclear burning to the variation of the temperature.

Let us start by introducing the concept of gravothermal specific heat. We shall not give much details, the reader may find a comprehensive discussion on this issue in Reference [40], whose approach we shall closely follow.

Thus let us consider a star in hydrostatic equilibrium, whose center is surrounded by a small sphere of radius $r_{s}$ and mass $m_{s}$. For a sufficiently small sphere, the central values of the pressure and the density, $P_{c}$, $\rho_{c}$, may be assumed to be equal to the pressure *P* at $r_{s}$ and the mean density in the sphere, respectively.

Let us now consider that a small amount of heat (dq) is added to the central sphere, producing its homologous expansion, then the following expression may be found (see Reference [40] for details):(38)$$dq=c_{P}T_{c}\left(\theta_{c}-\nabla_{ad}p_{c}\right)=c^{*}dT_{c},$$
where $c_{P}$ is the specific heat at constant pressure, $\nabla_{ad}$ is defined by Equation (Equation 24), $p_{c}$ and $\theta_{c}$ are defined by:(39)$$p_{c}\equiv \frac{dP_{c}}{P_{c}};\;\theta_{c}\equiv \frac{dT_{c}}{T_{c}},$$
and $c^{*}$, denotes the gravothermal specific heat, which relates the variations of heat and temperature and is defined by:(40)$$c^{*}\equiv c_{P}\left(1-\nabla_{ad}\frac{4\delta }{4\alpha -3}\right),$$
with
(41)$$\alpha \equiv {\left(\frac{\partial \ln\rho }{\partial \ln P}\right)}_{T};\;\delta \equiv -{\left(\frac{\partial \ln\rho }{\partial \ln T}\right)}_{P},$$
where the subscript *T* and *P* implies that the quantity is evaluated at constant temperature and pressure respectively.

If $c^{*}$ is positive (as for a nonrelativistic degenerate gas), any increasing of energy in the central sphere would warm up the matter, which eventually may lead to a thermal runaway. On the contrary if $c^{*}<0$ (as for an ideal monoatomic gas), any dq>0 produces a cooling (dT<0) reducing the overproduction of energy, avoiding thereby the thermal runaway (as it happens in the sun, fortunately!).

So far we have not specified the source of heat. Let us now consider the case when energy is generated by nuclear reactions and transported out of it by radiation in the diffusion approximation (assuming that the central region is not convective).

Then denoting by ϵ and $\lambda_{s}$ the mean energy generation rate (per unit mass) and the energy per unit time which leaves the sphere, respectively, the equilibrium condition reads:(42)$$\epsilon m_{s}-\lambda_{s}=0.$$

Next, let us now perturb (Equation 42) on a time scale which is short as compared to the relaxation time τ and the thermal adjustment time $\tau_{d}$ but much larger than the hydrostatic time scale.

Then using (Equation 38) and (Equation 42), we obtain for the energy balance of the perturbed state:(43)$$m_{s}d\epsilon -d\lambda_{s}=m_{s}\frac{dq}{dt}=m_{s}c^{*}\frac{dT_{c}}{dt}.$$

Next, it can be shown that, always keeping the homologous regime for simplicity, we may write (see Reference [40] for details):(44)$$\lambda_{s}^{-1}d\lambda_{s}=4x+4\theta_{c}-\kappa_{p}p_{c}-\kappa_{T}\theta_{c},$$
with dr=rx and $\kappa_{p}\equiv {\left(\frac{\partial \ln\kappa }{\partial \ln P}\right)}_{T}$, $\kappa_{T}\equiv {\left(\frac{\partial \ln\kappa }{\partial \ln T}\right)}_{P}$.

Then, introducing (Equation 44) into (Equation 43) we obtain:(45)$$m_{s}\lambda_{s}^{-1}\frac{dq}{dt}=m_{s}\lambda_{s}^{-1}c^{*}T_{c}\frac{d\theta_{c}}{dt}=\left[\epsilon_{T}+\kappa_{T}-4+\frac{4\delta }{4\alpha -3}\left(1+\epsilon_{p}+\kappa_{p}\right)\right]\theta_{c},$$
where $\epsilon_{p}\equiv {\left(\frac{\partial \ln\epsilon }{\partial \ln P}\right)}_{T}$ and $\epsilon_{T}\equiv {\left(\frac{\partial \ln\epsilon }{\partial \ln T}\right)}_{P}$.

The equation above is, formally, the same independently on the transport equation, however the explicit expression of the luminosity function $\lambda_{s}$ does depend on the transport equation to be used.

From a simple inspection of (Equation 45), it follows that whenever the square bracket in (Equation 45), $\lambda_{s}$ and $\theta_{c}$ are positive, then $\frac{dq}{dt}>0$. Thus, a positive sign of $c^{*}$ would produce a heat up ($\frac{d\theta_{c}}{dt}>0$) while a negative value of $c^{*}$ would produce a cool off ($\frac{d\theta_{c}}{dt}<0$). It is then obvious that the explicit expression of $\lambda_{s}$ is critical for the final verdict about the (in)stability of the nuclear burning in any scenario.

Now, the calculation of $\lambda_{s}$ proceeds as for the obtention of luminosity of the convective blob in the example analyzed in the previous subsection. Thus we may write as in (Equation 25)
(46)$$\lambda_{s}=S\left|\vec{q}\right|=\frac{2S}{d}\int_{-\infty }^{t}Q\left(t-t^{\prime }\right)DT\left(t^{\prime }\right)dt^{\prime }.$$
where S is the surface of the sphere and DT denotes the temperature difference between the central sphere and the surrounding matter.

Then, the evolution equation for DT is (Equation 27), that is,
(47)$$\frac{dDT}{dt}=-\frac{1}{\kappa \tau_{d}}\int_{-\infty }^{t}Q\left(t-t^{\prime }\right)DT\left(t^{\prime }\right)dt^{\prime }.$$

In the past the Maxwell–Fourier kernel has been used in (Equation 46) (e.g., Reference [40]), leading to a DT of the form of (Equation 29). However as mentioned before this might be inaccurate in problems where the order of magnitude of τ is larger or at least equal to the characteristic time scales of the system. That’s why such calculation was performed for the Cattaneo equation in Reference [47], where a DT of the form given by (Equation 33) is obtained. For our kernel (Equation 16) the form of DT is given by the Equation (Equation 34), whose solution describes a damped oscillatory behaviour as illustrated by (Equation 36).

In the light of previous results and comments we become aware of the relevance of processes occurring before relaxation. Indeed, let us focus on Equation (Equation 45). From standard results in nuclear physics, it can be assumed that the square bracket in (Equation 45) is positive. Therefore if $c^{*}>0$ any temperature increase would imply burning instability if $\lambda_{s}$ is positive. But as we have just seen, with our kernel, $\lambda_{s}$ exhibits an oscillatory behaviour, which means that before relaxation the system will jump from unstable to stable and vice-versa with a period depending on τ and $\tau_{d}$. In other words, for sufficiently large relaxation time, a quasi-periodic structure in the emission is expected to be observed before the system attains the stationary regime. Some astrophysical consequences of this fact have been discussed in detail in Reference [47].

### 3.3. The Relativistic Regime

So far we have considered fluids in the classical regime (non–relativistic), however many of the applications of the subject discussed here are expected to be used in the study of very compact objects where Newtonian gravity is no longer reliable. Accordingly, it would be wise to consider the relativistic generalization of the presented model.

Relativistic causal dissipative theories has been the subject of many research works in the past. Particularly relevant are References [48,49,50,51,52]. However, curiously enough, in the classical limit the transport equations proposed by these authors lead to the Cattaneo–Vernotte equation. Therefore it is justified to ask what is the relativistic equation producing (Equation 19) in the non–relativistic limit.

The relativistic transport equation in the Israel–Stewart theory reads
(48)$$\tau h_{\nu }^{\mu }q_{;\beta }^{\nu }V^{\beta }+q^{\mu }=-\kappa h^{\mu \nu }\left(T_{,\nu }+Ta_{\nu }\right)-\frac{1}{2}\kappa T^{2}{\left(\frac{\tau V^{\alpha }}{\kappa T^{2}}\right)}_{;\alpha }q^{\mu },$$
where $V^{\beta }$, $h_{\nu }^{\mu }$, $q^{\mu }$, $a_{\nu }$ denote the four–velocity, the projector on the hypersurface orthogonal to the four–velocity, the four–heat flux vector and the four–acceleration respectively. The semicolon denotes covariant derivative and Greek indices run from 0 to 4, with 0 corresponding to the timelike component, whereas 1,2,3 correspond to the spatial components.

Let us analyze in some detail the different terms entering in (Equation 48), and their non–relativistic limit. For doing that we have to keep in mind that we are considering co-moving observers, implying $V^{\alpha }=\left(\frac{1}{\sqrt{|g_{tt}|}},0,0,0\right)$ where $g_{tt}$ denotes the tt component of the metric tensor and we are using relativistic units implying that the light velocity and the Newtonian gravitational constant equal to one.

The first term on the left is the “hyperbolizer” term and in the non–relativistic limit becomes $\tau \frac{\partial \vec{q}}{\partial t}$. The second term on the left is just the heat flux vector, becoming $\vec{q}$ in the non–relativistic regime. Obviously these two terms must be present in any causal theory of dissipation. Next, the first term on the right of (Equation 48) ($\kappa h^{\mu \nu }T_{,\nu }$) is also trivially identified as the temperature gradient term in the non–relativistic limit, on the contrary the second term on the right ($Ta_{\nu }$) is purely relativistic. It was originally discovered by Tolman [53] when investigating the conditions of thermal equilibrium in a gravitational field and reflects the fact that a temperature gradient is necessary to ensure the thermal equilibrium, due to the inertia of heat. In the non–relativistic regime this term vanishes but it must be present in any relativistic dissipative theory.

Thus, the first, the second and the fourth terms appearing in (Equation 48) are expected to be present in any relativistic causal theory of dissipation. Instead, the last term in (Equation 48) is characteristic of the Israel–Stewart theory and is absent in some other theories, (e.g., Reference [54]). It should be mentioned that in some relativistic theories additional terms may be added which vanish in the classical limit.

Therefore, as mentioned before, the non–relativistic limit of (Equation 48) is just the Cattaneo–Vernotte equation and we have to propose something different to (Equation 48) in order to recover (Equation 18) in the non–relativistic regime.

From the above comments we see that the options are restricted to changes in the third term. The simplest modification of (Equation 48) leading to (Equation 18) in the non–relativistic limit, reads:(49)$$\tau h_{\nu }^{\mu }q_{;\beta }^{\nu }V^{\beta }+q^{\mu }=-\kappa h^{\mu \nu }\int_{-\infty }^{t}T_{,\nu }\sqrt{|g_{tt}|}dt^{\prime }-\kappa Ta^{\mu }-\frac{1}{2}\kappa T^{2}{\left(\frac{\tau V^{\alpha }}{\kappa T^{2}}\right)}_{;\alpha }q^{\mu }.$$

It goes without saying that, for any specific problem, using (Equation 49) instead of (Equation 48) would lead to quite different conclusions. However any of such application is out of the scope of this work and we only wish to stress here the relevance of this point.

## 4. Conclusions

We have put forward a new model of heat causal conduction not satisfying the fading memory paradigm. Although we are well aware of the fact that an almost infinite set of kernels violating the fading memory paradigm may be conceived, we wished here to propose a specific kernel in order to bring out the effects of this type of kernel in the two examples analyzed. Besides, it is worth mentioning that our kernel somehow represents the extreme case of violation of the fading memory paradigm.

Two specific applications were discussed in some detail to illustrate the critical dependence of the physical analysis of each scenario on the specific transport equation involved. The presented analysis clearly exhibits the observational differences appearing as the result of using our kernel instead of Maxwell–Fourier or Cattaneo. However much more detailed setups are required to propose specific experiments to confront those alternatives. In the same line of arguments it would be interesting to study different solutions to (Equation 21) under a variety of circumstances and compare the resulting temperature profiles with those obtained from the integration of (Equation 6).

For the sake of completeness we have also proposed a simple generalization of our model to the relativistic regime. Further applications of the resulting equation to specific problems involving relativistic dissipative processes are required to deduce the possible observational implications of the model.

The problem under consideration here may be approached by means of different frameworks, an example of which is the rational extended thermodynamics. It would be interesting to explore if new aspects of the proposed model could be brought out using any of these theories.

Finally, it is worth dedicating some thoughts about the possible situations where we could expect a dissipative process not complying with the fading memory paradigm. Obviously, for sufficiently large relaxation times the impact of the temperature gradients at the neighborhood of the observation time, on the observed heat flux, may be negligible. In this sense our kernel (Equation 16), as mentioned above, represents an extreme situation. Constitutive equations for the internal energy different from (Equation 3), could also lead to the same kind of kernel.

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
