# Peer review of "Causal Heat Conduction Contravening the Fading Memory Paradigm"

_entropy, 2019, doi:10.3390/e21100950_

Round 1

Reviewer 1 Report

The manuscript proposes a model of causal heat conduction that explicitly violates the fading memory paradigm. Overall, I think that this work displays some interesting and novel ideas and surely deserves to be published after some minor revisions.

First, it is worth mentioning that the all story of fading memory and causal heat propagation has attracted some attention, in the last couple of years, in the community of fractional calculus. Particularly in relation to the fractional Cattaneo-Maxwell equation. Ideally, the author could spend a few words on this topic in the introduction.

Second, I am wondering about the potential connection of the proposed model with the Rational Extended Thermodynamics. I would appreciate a few comments on this point in the conclusions.

Finally, aside from a few typos that can be found throughout the paper, the manuscript is generally well written. However, I believe that statements like "to throw out the baby with the bath water" and "a kind of Alzheimer memory behaviour" are beneath the level that can be tolerated in a scientific journal. Therefore, I invite the author to refrain form using these phrases.   

Author Response

See the attached cover letter

Reviewer 2 Report

In the manuscript “Causal heat conduction contravening the fading memory paradigm”, the author addresses the details of dissipative processes. 

He exposes with remarkably clarity the fundamental physical ideas and options we face when considering the transport equation underlying those processes. He provides a  comprehensive and rather concise review of the subject, in order to guide the reader to the novel approach that he puts forward regarding  the so-called fading memory paradigm. 

Indeed he introduces  a novel and alternative model which totally contravenes the latter paradigm,  where the dissipative system is essentially influenced by its remote past states. His proposal is illustrated by two interesting examples where such a novel viewpoint finds application: Thermoaline convection, and nuclear burning instability. These applications are particularly useful to render clear not only the impact of the  hypothesis violating the fading memory paradigm, but also to persuade us that there examples in  nature for which it contributes an adequate explanation.

Finally the author considers the relativistic generalization of his model, although  briefly. But this is understandable as a more detailed analysis of the modification of the Israel-Stewart transport equation is, presumably,  deserving of a follow-up work. 

So I believe that the present work is very interesting, and contributes a novel approach to dissipative processes, therefore deserving publication in the Entropy journal.

PS: Although the manuscript is rather well written, and presented, I spotted two minor typos in page 2, in lines 58 and 59 (fifth and fourth lines from the bottom of the page, respectively). In line 58 “lower that unity” should rather be “lower than unity”, and in line 59, “ the material cannot longer” should be replaced by “the material can no longer”.

Reviewer 3 Report

See the attachment.

Author Response

See the attached cover letter

This manuscript is a resubmission of an earlier submission. The following is a list of the peer review reports and author responses from that submission.

Round 1

Reviewer 1 Report

The Author does not present enough novelty for the study. What exactly the problem was solved? Author should more clarify their original contribution. There is no net information about the conclusion in the abstract. The abstract is very poor.

In my opinion, manuscript does not bring a lot of scientific values. No reliable conclusions have been submitted. Unfortunately, the study seems like the report of an unfinished study. In my opinion the manuscript is far off the level required by an international journal.

Reviewer 2 Report

The present paper deals with strongly nonlocal theories, discussing various kernels and their physical relevance. 
It is submitted into the SI: "Entropy Generation and Heat Transfer II".
However, according to the description of the Special Issue, this manuscript seems to be far from that topic.
Therefore, the Reviewer recommends to
1, extend the paper with much more thermodynamical discussion, e.g., ca. 8 state equations are used but the notion "equation of state" is not mentioned at all.
2, include the entropy production analysis into the discussion. That approach would clearly highlight the thermodynamic constraints regarding to the kernels and the coefficients within.
Although the topic itself is interesting, it suffers from several shortcomings.
The above suggestions will lead to a significant reformulation of the manuscript and it is suggested to the Author to consider to withdraw and resubmit the new paper.
Furthermore, it should be fitted to the topic SI.
Moreover, there are some further issues that must be considered for corrections or reformulation.
3, The language and style must be corrected. For example, the sentence "...transient phenomena affect the future of the system..." is trivial.
4, Fourier's law is often referred as "Maxwell-Fourier" law. This is completely misleading since Fourier was the first one (much earlier) in this regard.
Moreover, according to the convention of the related literature, it is "Fourier's law", not "Maxwell-Fourier" law.
5, The balance equations must be mentioned at the beginning, not on the 5th page first.
The present discussion entirely leaves the role of thermodynamics out of sight. Thus, it is strongly recommended to appropriately introduce the constitutive equations (such as Fourier's law).
It is especially important when the Author starts to apply a series expansion on a constitutive equation.
This is strictly prohibited by the second law of thermodynamic, and that mistake becomes apparent on the example of dual phase lag equations.
This is the reason why the entropy production must be calculated for each model.
Or, at least, the possible problems must be mentioned.
6, The notions of "causality" and "hyperbolicity" are not the same. It should be highlighted that even for the relativistic case, the propagation speed can be higher than the speed of light (theoretically possible).
There is no upper limit in the presented hyperbolic theories. Moreover, the parabolic Guyer-Krumhansl equation is successfully applied in low-temperature experiments since it predicted the so-called window condition that was vital to find "second sound" in solids.
7, On page 2, the Author states that the relaxation time is the quantity that is needed to return the system into steady state.
It does not make sense in that form, as it is not true. For instance, even for especially high relaxation times (~1 s), it does not measure the time to return to the steady state.
In the same paragraph, it is stated that the relaxation time is not connected to the mean collision time of the particles.
It is stronly recommended to discuss it more deeply and prove it with calculations, especially in the light of kinetic theory.
Otherwise, it remains only a "hypothesis" without any reasoning. More importantly, it seems that it is exploited later in the construction of kernels, and that would modify the outcome of the results.
8, The origin of eq. (6) is not clear for a general reader. What does it mean that a constitutive equation (3) is "solved for q"? (3) itself is an evolution equation (a constitutive one) for q. Why is it called "solution"?
9, On page 3, it is stated that the memory effect is related to gradient of T, instead of q. It must be discussed more deeply in the light of Extended Irreversible Thermodynamics in which the heat flux has a memory (as its time derivative appears instead of \partial_t \nabla T).
10, Eq. (11) is not a valid model for heat conduction, according to the second law, e.g., see the Green-Naghdi theory.
11, The solution of eq. (14) would be interesting to be discussed, especially emphasizing the role of initial conditions as the lower limit is -\infty. That history of temperature gradient is completely unknown.
12, On page 5, it is mentioned that the outcome for temperature is not unique and it depends on the balance equation.
It must be presented through calculations, emphasizing the role of balance equation and the equation of state for internal energy (eq. (18)).
13, On page 6, the nabla operator-like notation is used to denote state equations. So, first, it is suggested to use a different notation. Second, discuss the thermodynamical relation regarding the state equations, e.g., which are presented in eq. (23).
14, The origin of eqs. (24-25) is not clear. The related calculation must be presented, since V depends on DT. What is "d" here? How does the radiation come into the picture?
15, Eq. (30) is seemingly not thermodynamically compatible for the temperature. It seems that Fourier's law does not produce entropy which would be weird.
16, Perturbation of (41) is not clear.
17, Later, 6 more EoS are introduced without any thermodynamical discussion. It is recommended to include.
18, The relativistic part does not play an important role in the manuscript. Moreover, simply introducing relativistic equation based on some analogy is not possible using no strict thermodynamical background.
As it is completely missing, that part must be reformulated.
19, In the conclusion, the validation of these theories is mentioned. There are numerous experimental results in the literature. Thus it is recommended to solve the presented models for these realistic problems and compare them in a different paper.